# Dynamic plasticity in phototransduction regulates seasonal changes in color perception

Tsuyoshi Shimmura[1,2,3,12], Tomoya Nakayama [1,3], Ai Shinomiya[1,2], Shoji Fukamachi[4], Masaki Yasugi[5], Eiji Watanabe[2,5], Takayuki Shimo[1,3], Takumi Senga[1,3], Toshiya Nishimura[6,7], Minoru Tanaka[2,6,7], Yasuhiro Kamei [2,8], Kiyoshi Naruse [2,9] & Takashi Yoshimura [1,3,10,11]

To cope with seasonal changes in the environment, organisms adapt their physiology and behavior. Although color perception varies among seasons, the underlying molecular basis and its physiological significance remain unclear. Here we show that dynamic plasticity in phototransduction regulates seasonal changes in color perception in medaka fish. Medaka are active and exhibit clear phototaxis in conditions simulating summer, but remain at the bottom of the tank and fail to exhibit phototaxis in conditions simulating winter. Mate preference tests using virtual fish created with computer graphics demonstrate that medaka are more attracted to orange-red-colored model fish in summer than in winter. Transcriptome analysis of the eye reveals dynamic seasonal changes in the expression of genes encoding photopigments and their downstream pathways. Behavioral analysis of photopigment-null fish shows significant differences from wild type, suggesting that plasticity in color perception is crucial for the emergence of seasonally regulated behaviors.

[1] Division of Seasonal Biology, National Institute for Basic Biology, National Institutes of Natural Sciences, Okazaki, Aichi 444-8585, Japan. [2] Department of Basic Biology, The Graduate University for Advanced Studies (SOKENDAI), Hayama 240-0193, Japan. [3] Laboratory of Animal Physiology, Graduate School of Bioagricultural Sciences, Nagoya University, Nagoya, Aichi 464-8601, Japan. [4] Department of Chemical and Biological Sciences, Japan Women's University, Bunkyo-ku, Tokyo 112-8681, Japan. [5] Laboratory of Neurophysiology, National Institute for Basic Biology, National Institutes of Natural Sciences, Okazaki, Aichi 444-8585, Japan. [6] Laboratory of Molecular Genetics for Reproduction, National Institute for Basic Biology, Okazaki, Aichi 444-8787, Japan. [7] Division of Biological Science, Graduate School of Science, Nagoya University, Nagoya, Aichi 464-8601, Japan. [8] Spectrography and Bioimaging Facility, National Institute for Basic Biology, National Institutes of Natural Sciences, Okazaki, Aichi 444-8585, Japan. [9] Laboratory of Bioresources, National Institute for Basic Biology, National Institutes of Natural Sciences, Okazaki, Aichi 444-8585, Japan. [10] Avian Bioscience Research Center, Graduate School of Bioagricultural Sciences, Nagoya University, Nagoya, Aichi 464-8601, Japan. [11] Institute of Transformative Bio-Molecules (WPI-ITbM), Nagoya University, Nagoya, 464-8601 Aichi, Japan. [12] Present address: Department of Biological Production, Tokyo University of Agriculture and Technology, Fuchu, Tokyo 183-8509, Japan. Tsuyoshi Shimmura and Tomoya Nakayama contributed equally to this work.  Correspondence and requests for materials should be addressed to T.Y. (email: takashiy@nibb.ac.jp)

Seasonal changes in color sensitivity could underlie seasonal behavioral changes. For examples, a recent study in humans demonstrated that the wavelength settings for the "unique yellow hue" are significantly shifted to shorter wavelengths in summer compared with those in winter[1, 2]. Seasonal affective disorder patients, experiencing recurrent winter episodes of depressed mood, overeating and hypersomnia[3], show electroretinogram changes in winter, with lower sensitivity compared with healthy subjects[4]. This decreased retinal sensitivity, along with the depressed mood, are normalized following light therapy or in summer[4]. These studies highlight the potential importance of the retina in seasonality, but the molecular basis of these seasonal changes remains unknown.

There is also evidence in non-human animals that opsin expression varies in response to environmental changes including seasons, for example, (i) SWS1 (short-wavelength sensitive; ultraviolet sensitive) opsin expression in damsel fish changes in winter compared with those in summer[5]; (ii) LWS (long-wavelength sensitive; red sensitive) opsin in stickleback[6] and OPN4 (opsin4; melanopsin)-related genes in zebrafish[7] vary in response to photoperiod; (iii) LWS and RH2 (rhodopsin 2; green sensitive) opsins are differentially expressed according to water depth in damsel fish[5]; (iv) SWS2B (blue/violet) opsin changes in wild-caught and lab-reared cichlids[8]; and (v) coexpression of LWS and RH2A is influenced by environmental background spectra in cichlid[9]. Although these gene expression changes appear to cause alterations in the spectral sensitivity of vision, their physiological and ecological significance remains unclear.

In this study, we addressed the functional significance of seasonal changes in opsin expression directly using Japanese medaka (*Oryzias latipes*), an excellent model for studying seasonal adaptation[10]. We first characterized seasonal changes in behavior and found differences in phototaxis and mate preference. We then examined the impact of these changes (that is, in response to photoperiod and temperature) on global gene expression within the eye and found dynamic changes in the expression of genes encoding photopigments and their downstream pathways. Finally, we demonstrated the functional significance of seasonally regulated plasticity in opsin gene expression on seasonally regulated behaviors, using fish harboring null mutations in the LWS opsins, and found that seasonal changes in color sensitivity underlie seasonal behavioral changes.

## Results

**Seasonal changes in behavioral traits.** We observed seasonal differences in behaviors in a medaka population maintained in outdoor enclosures for several years. To confirm this observation in a controlled environment, we tracked the spatial usage of two fish (male and female) in a home tank using a three-dimensional (3D) tracking system. Similar to their activity when reared outdoors, fish kept under long-day (LD), warm-temperature conditions (LW: 14 h light/10 h dark; 26 °C) swam all over the tank, whereas fish kept under short-day (SD) and cool-temperature conditions (SC: 10 h light/14 h dark; 8 °C) stayed at the bottom of the tank (Fig. 1a). In general, fish swim much faster and further in LW compared with SC conditions (Supplementary Fig. 1a). Fish normally swim toward a weak light stimulus (positive phototaxis), but avoid strong light signals (negative phototaxis)[11]. They also exhibit a transient period of hyperactivity upon loss of illumination, known as darkness-induced light-seeking behavior[12]. In this study, we observed negative phototaxis and darkness-induced light-seeking behavior in medaka kept under LW conditions. When LW medaka were introduced into a novel tank (see Methods section), they avoided white light (23.1 μmol m$^{-2}$ s$^{-1}$; 70 Lux) (Supplementary Fig. 2a),

but immediately moved to the opposite side of the tank just after lights out (Fig. 1b). By contrast, SC fish swam randomly before and after lights-out and failed to exhibit either negative phototaxis or darkness-induced light-seeking behavior (Fig. 1b). It is noteworthy that although overall activity is decreased under winter conditions in the home tank (Fig. 1a and Supplementary Fig. 1a), these fish have the capacity to move when transferred to new environmental conditions (e.g., test tank) (Supplementary Fig. 1b)

During the breeding season, medaka develop several black stripes on the caudal fin and black spots on the ventral fins. In addition, the orange–red color along the dorsal and ventral margins of the caudal fin becomes more intense (Supplementary Fig. 2b). This nuptial coloration is caused by an increase in the number of melanophores and xanthophores[13]. Therefore, we hypothesized that medaka might be attracted to orange–red-colored mates during the breeding season. To test this hypothesis, we conducted a mate preference test using virtual fish generated with 3D computer graphics (3D-CG) (Fig. 1c and Supplementary Movie 1). In this system, different colored 3D fish (Supplementary Fig. 2c) with realistic motion patterns can move around on the screen, enabling evaluation of the effect of body color independently of behavioral patterns or chemical signals of attraction (e.g., pheromones). Neither LW nor SC fish showed a preference for gray model fish (Supplementary Fig. 2c). However, LW fish spent more time on the screen side of the tank when presented with orange–red-colored model fish. By contrast, SC fish did not exhibit such a preference (Fig. 1d). Based on these results, we speculate that medaka kept under LW and SC conditions process the visual environment differently.

**Seasonal regulation of genes involved in phototransduction.** To address the mechanism of these visual changes, we performed genome-wide expression analysis using microarrays to examine seasonal changes in global gene expression within the eye. As seasonal responses in gonadal size are much more robust in females than in males, we used female medaka for the microarray analysis. When SC fish were transferred to LW conditions, a significant increase in the gonad somatic index (gonadal weight/body weight × 100) was first detected on day 3 and ovaries reached their full size on day 7 (Supplementary Fig. 3). Eyes were collected from six fish at 16 h after light onset on days 0, 1, 2, 3, 5, 7 and 14 after transfer from SC to LW conditions. For each time point, biotinylated complementary RNAs (cRNAs) prepared from pooled eyes (n = 3) were hybridized to duplicate sets of microarrays to minimize experimental error. We observed no fluctuation in the expression of ribosomal protein L7 (RPL7), an internal RNA control[14], during the transition (Supplementary Fig. 4a). The results of the microarray experiment were validated by quantitative PCR (qPCR) assays in both male and female fish (Supplementary Fig. 5).

Differential gene expression was subjected to one-way analysis of variance (ANOVA) analysis (P < 0.01, Tukey's HSD (Honest Significant Difference) post-hoc test and a Benjamini–Hochberg FDR (False Discovery Rate) P < 0.001) with a twofold cut-off for fold change. This analysis identified 824 LW-induced and 727 LW-suppressed genes (Supplementary Data 1). Among these, seven opsin genes (*SWS1*, *RH2-A*, *RH2-B*, *LWS*, *RH1*, (*rhodopsin 1*), *OPN4L* and *OPN5L* (*opsin 5 like*)) were upregulated, whereas two opsins (*parapinopsin*: *PP* and *peropsin*: *RRH*) were downregulated (Fig. 2a and Supplementary Figs 4 and 5). Among the opsin genes, *LWS-A* and *LWS-B* are highly similar, with 98.8% identity[15]. Consequently, it is impossible to distinguish *LWS-A* and *LWS-B* by microarray analysis or in situ hybridization. Therefore, we designated *LWS-A* and *LWS-B* as *LWS* in this study.

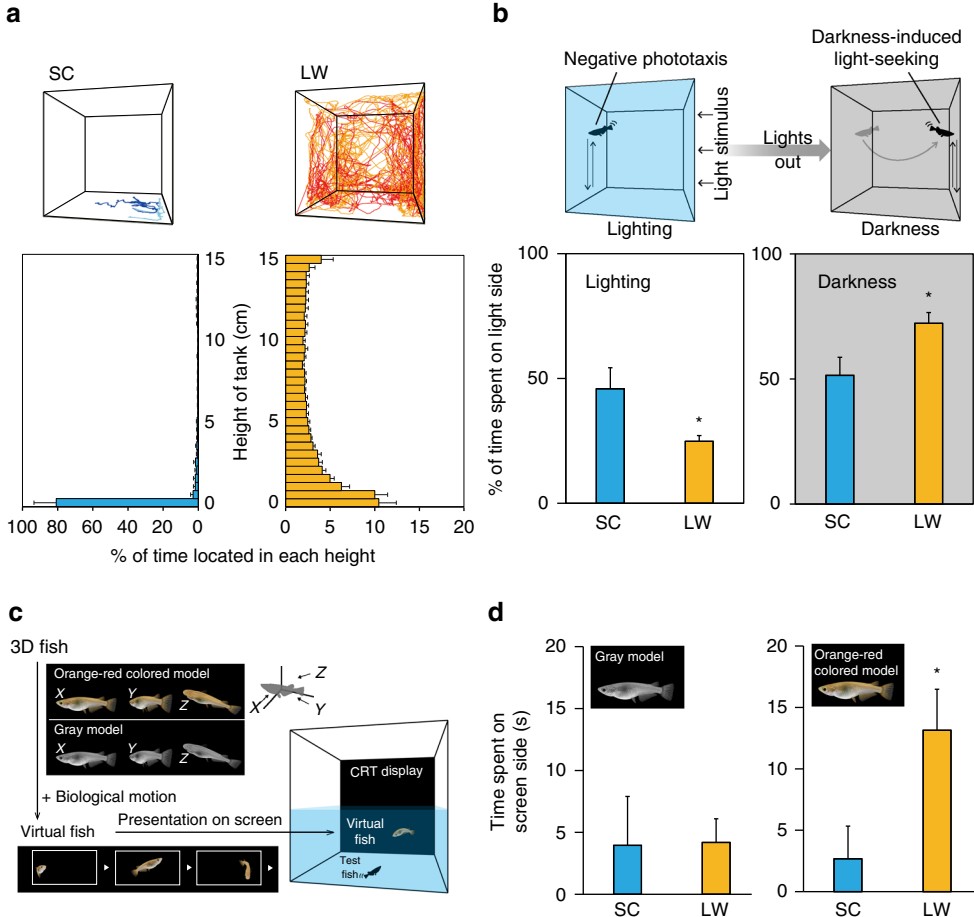

**Fig. 1** Medaka undergo seasonal changes in behavior. **a** *Top*: Behavioral traces of two individuals under long day/warm temperature (LW) and short day/cool temperature (SC) conditions. Different colors denote traces for each individual. *Bottom*: Time spent at each height of the tank. **b** *Top*: Medaka (LW fish) exhibited negative phototaxis (*left*) and darkness-induced light-seeking behavior (*right*). *Bottom*: (*left*) LW fish exhibited negative phototaxis to a white light stimulus, whereas SC fish swam randomly (*$P < 0.05$, $t$-test; mean + SEM, $n = 5$–9). (*right*) When the light was turned off, LW fish immediately moved to the opposite side of the tank, whereas SC fish failed to exhibit this darkness-induced light-seeking behavior (*$P < 0.05$, $t$-test; mean + SEM, $n = 5$–9). **c** Three-dimensional computer graphics (3D-CG) medaka (see Supplementary Movie 1). **d** (*left*) When presented with a gray model fish on the screen, neither SC nor LW test fish showed a preference ($P > 0.05$, $t$-test; mean + SEM, $n = 8$–10). (*right*) By contrast, LW test fish spent more time on the screen side when presented with orange-red-colored model fish, whereas SC fish did not show a preference (*$P < 0.05$, $t$-test; mean + SEM, $n = 10$)

In addition to the opsin genes, we observed gene expression changes in their downstream phototransduction pathways[16–18]. Rhodopsin and cone opsins couple with the G-protein transducin (rods: GNAT1, GNB1 and GNGT1; cones: GNAT2, GNB3 and GNGT2). We observed upregulation of these genes by the LW stimulus (Fig. 2b and Supplementary Figs. 4 and 5). Transducin activates phosphodiesterase (PDE6) and we observed upregulation of *PDE6G* and downregulation of *PDE6GL* (Fig. 2b and Supplementary Figs. 4 and 5). PDE6 catalyzes the conversion of cGMP to 5'-GMP, and cGMP acts on cyclic nucleotide-gated (CNG) channels (rods: CNGA1 and CNGB1; cones: CNGA3 and CNGB3). The LW stimulus resulted in upregulation of *CNGB1A* and downregulation of *CNGB3* (Fig. 2b and Supplementary Figs. 4 and 5). cGMP synthesis is mediated by guanylate cyclase (GC) and GC activity is regulated by $Ca^{2+}$ and GC-activating protein (GCAP). *GCAP3* was suppressed by the LW stimulus (Fig. 2b and Supplementary Figs. 4 and 5). Activated opsin is inactivated by rhodopsin kinases (RK: GRK1 for rods and GRK7 for cones) and arrestin (Arr). LW induced upregulation of *GRK1A* and suppression of *GRK1* and *GRK7* (Fig. 2b and Supplementary Figs. 4 and 5). RK-mediated phosphorylation of activated opsin is regulated by recoverin (RCVRN). Three genes that encode RCVRNs were all induced by the LW stimulus (Fig. 2b and Supplementary Figs. 4 and 5). Light converts 11-*cis* retinal to all-*trans* retinal and in order for photoreceptors to function all-*trans* retinal must be converted back to 11-*cis* retinal. Among various retinol dehydrogenases (RDHs), RDH8 is largely responsible for this activity[19] and *RDH8* was induced by LW (Fig. 2b and Supplementary Figs. 4 and 5). Optical plasticity in the crystalline lens has been reported in fish[20]. Interestingly, we also observed LW induction of genes involved in lens formation, such as *MAF2*, *LIM2*, *CRYBB2* and *CRYBB1L3* (Supplementary Fig. 4).

This large family of opsin proteins spans a wide range of wavelengths, from ultraviolet to far red[15] (Supplementary Fig. 2d). Owing to the large number of opsins we identified in our microarray analysis and the overlap in their spectral sensitivities, functional characterization of each opsin would be complicated and challenging. We therefore chose to focus on LWS for further functional analysis, because our behavioral studies showed medaka preferred orange–red-colored fish (Fig. 1d). In situ hybridization analysis confirmed LW induction of *LWS* in the photoreceptor layer in the whole retina (Fig. 2c and Supplementary Fig. 6).

**LWS-null fish fail to exhibit a summer phenotype.** LWS-null ($LWS^{-/-}$) medaka generated using the CRISPR/Cas9 system were unable to perceive red light[21]. To test the effects of LW-induced LWS on negative phototaxis, we examined the effect of white (23.1 µmol m$^{-2}$ s$^{-1}$; 70 Lux) and red light ($\lambda_{max} = 730$ nm, half-bandwidth = 21 nm, 13.3 µmol m$^{-2}$ s$^{-1}$) (Supplementary Fig. 2a) on wild-type and LWS-null fish kept under LW conditions. It is noteworthy that although the $\lambda_{max}$ of LWS is ~560 nm, wild-type medaka have been shown to respond behaviorally to light at unexpectedly long wavelengths (up to 830 nm) under light-adapted conditions[21]. In contrast, LWS-null fish could respond only up to 740 nm, demonstrating that no other cone opsins except LWS could be responsible for absorbing light at wavelengths >740 nm. Therefore, monochromatic red light at 730 nm should be readily detected by the wild-type medaka, whereas LWS-null medaka are much less sensitive to this wavelength and should have difficulty detecting this light. Wild-type fish exhibited negative phototaxis in response to both white and red light (Fig. 3a). As white light contains long wavelengths of light (Supplementary Fig. 2a), LWS-null fish exhibited less negative phototaxis in response to white light (Fig. 3a, left), suggesting reduced photosensitivity in LWS-null fish. Critically, LWS-null fish exhibited darkness-induced light-seeking behavior following the switch to monochromatic red light (Fig. 3a, right), demonstrating that LW induction of LWS is required for negative phototaxis to red light. We also performed mate preference tests using the 3D-CG medaka. Consistent with the results shown in Fig. 1d, wild-type fish kept under LW conditions preferred orange–red-colored fish, but not gray fish (Fig. 3b). However, although LWS-null fish are fertile under LW conditions, their behavior was clearly affected, as they exhibited a weaker preference for orange–red-colored model fish than wild-type fish (Fig. 3b, right). Furthermore, there was no difference in the preference of LWS-null fish to either orange-red-colored or gray model fish ($t = 1.3$, d.f. = 26, $P = 0.21$, t-test). Thus, our results demonstrated that LW induction of LWS contributes to LW-induced mate preference.

**LWS expression is not mediated by hormones or photoperiod.** Changes in opsin expression by environmental stimuli have recently been reported in several species[5–9], but the mechanism of these changes is not well understood. A recent study in stickleback showed that testosterone increases LWS expression[6]. We next tested whether seasonal sex hormones influence LWS expression in medaka. Sex hormone is predominantly secreted under LW conditions (i.e., breeding season), but not under SC conditions (i.e., non-breeding season). First, castration of males did not affect LWS expression under LW conditions (Fig. 4b). We then examined the effect of testosterone (T), 17α-methyltestosterone (MT) and 17β-estradiol (E2) under SC conditions, and none of these treatments affected LWS expression (Fig. 4c). Finally, we tested the effect of cool and warm temperatures in combination with SD and LD conditions and discovered that warm temperature induced the expression of LWS (Fig. 4d). These results suggest that temperature is the primary factor regulating LWS in medaka and not testosterone or light.

## Discussion

We demonstrated that seasonal plasticity in light and color perception underlies seasonally regulated behaviors in medaka. Our study demonstrated the seasonal regulation of photopigments and their downstream phototransduction pathways. These seasonal changes appear to modulate light sensitivity and the rates of recovery and light/dark adaptation. In winter, both day length and light intensity decrease and medaka are less active and stay on the riverbed. As medaka rarely eat in winter, we predict that they need to save energy. We speculate from our results that medaka could be saving energetically costly visual functions during winter by downregulating the expression of genes involved in phototransduction. By contrast, light intensity is high during spring and summer. Animals have evolved body coloration, ornamentation and nuptial coloration to attract mates[22]. These dynamic changes in body color and appearance emphasize the importance of vision in seasonal breeding. The effect of

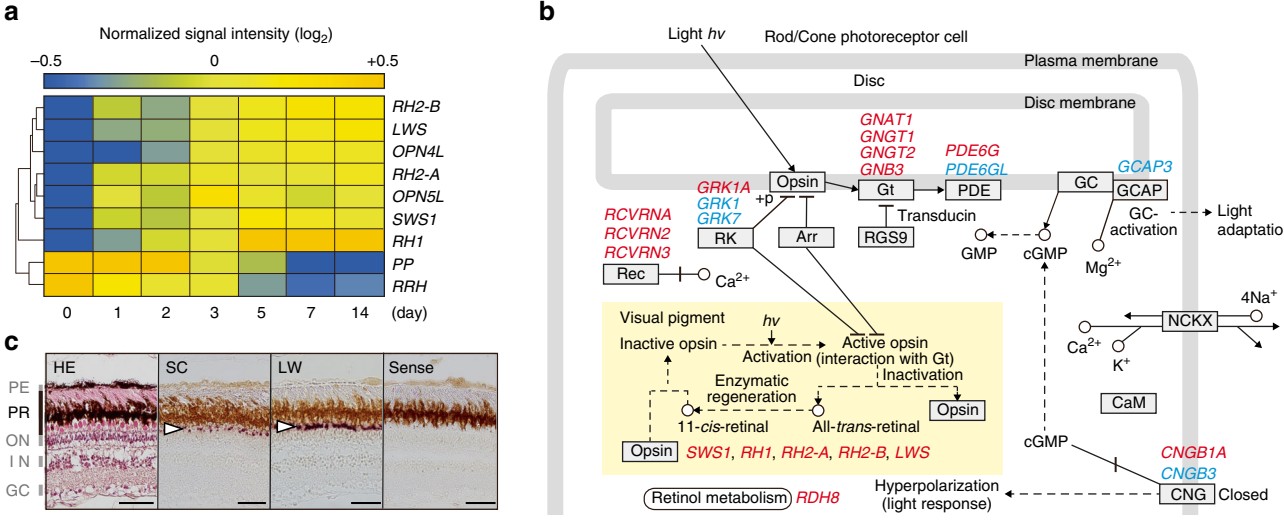

**Fig. 2** Genome-wide expression analysis of the eye reveals dynamic seasonal changes in phototransduction. **a** Clustered organization of seven upregulated and two downregulated opsin genes following a long day/warm temperature (LW) stimulus. Data were normalized over the complete data set. The color scale represents the normalized signal intensity. **b** Seasonal changes in the expression of genes encoding photopigments and their downstream phototransduction pathways. Gene symbols in *red* and *blue* indicate up- and downregulated genes, respectively. Adapted from the KEGG (Kyoto Encyclopedia of Genes and Genomes) phototransduction pathway. **c** In situ hybridization analysis confirmed LW induction of *LWS* opsin in the photoreceptor layer (*GC* ganglion cell layer, *IN* inner nuclear layer, *ON* outer nuclear layer, *PE* pigment epithelium, *PR* photoreceptor layer). Representative images from the central area of the retina (also see Supplementary Fig. 6). *White arrow* indicates the hybridization signal. *Scale bar*: 50 µm

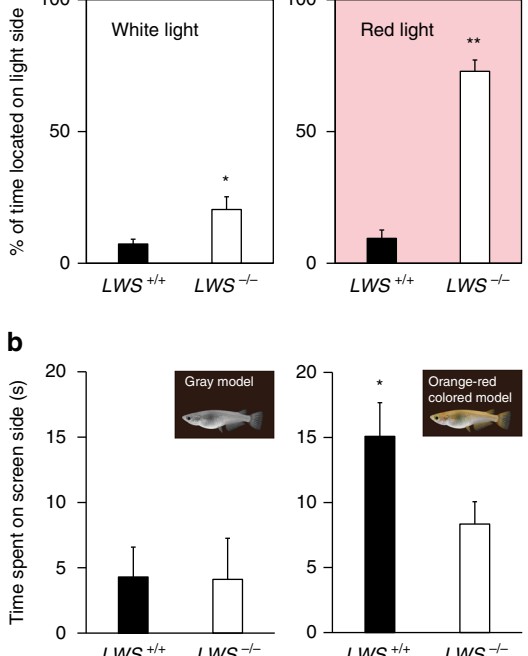

**Fig. 3** *LWS* opsin-null fish exhibit defects in phototaxis and mate preference. **a** *LWS*$^{-/-}$ fish exhibited impaired negative phototaxis to white light compared to *LWS*$^{+/+}$ fish under long day/warm temperature (LW) conditions (**P* < 0.05, *t*-test; mean + SEM, *n* = 6–12). When white light was switched to red light, *LWS*$^{-/-}$ fish exhibited light-seeking behavior, whereas *LWS*$^{+/+}$ fish continued to exhibit negative phototaxis (***P* < 0.01, *t*-test; mean + SEM, *n* = 6–12). **b** Neither *LWS*$^{+/+}$ nor *LWS*$^{-/-}$ fish kept under LW conditions showed a preference for *gray* model fish (**P* > 0.05, *t*-test; mean + SEM, *n* = 10–11). *LWS*$^{+/+}$ fish preferred orange-red-colored model fish, but this preference was impaired in *LWS*$^{-/-}$ fish (*P* < 0.05, *t*-test; mean + SEM, *n* = 16–18)

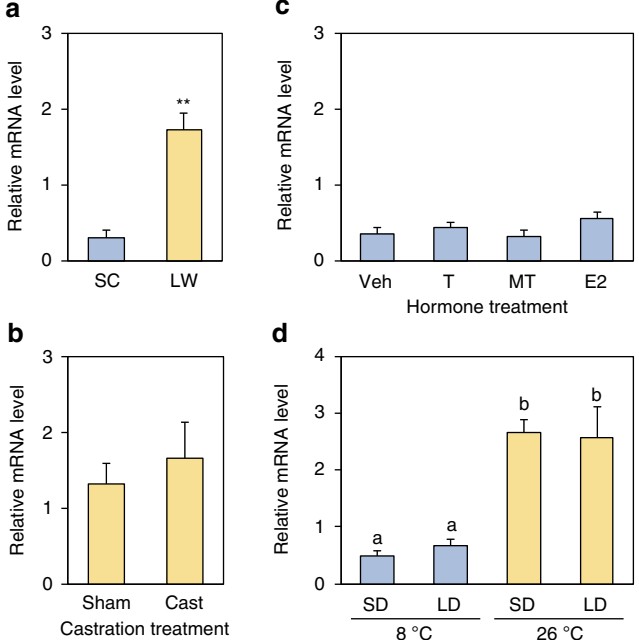

**Fig. 4** Regulation of the *LWS* opsin gene by temperature. **a** LW (long day/ warm temperature)-induction of *LWS* expression in males (***P* < 0.01, *t*-test; mean + SEM, *n* = 5). **b** Effects of castration (Cast) on *LWS* expression under LW conditions. (*P* > 0.05, *t*-test; mean + SEM, *n* = 4–5). **c** Effect of testosterone (T), methyltestosterone (MT) and estradiol (E2) on *LWS* expression under SC conditions ($F_{3,16}$ = 1.53, *P* > 0.05, ANOVA; mean + SEM, *n* = 5). Veh: vehicle. **d** Effect of temperature in combination with short-day (SD) and long-day (LD) conditions on *LWS* expression. Different characters indicate significant differences ($F_{3,16}$ = 14.1, *P* < 0.01, ANOVA; Fisher's least significant difference (LSD) test; mean + SEM, *n* = 5)

environmental changes on opsin gene expression have been recently reported in several teleosts[5–9]. Therefore, seasonal plasticity in color perception is most likely not specific to medaka, but rather a potentially common mechanism shared by many species. Seasonal plasticity has also been reported to occur in the auditory system[23, 24]. Specifically, plasticity in ion channel expression maximizes the detection of conspecific courtship auditory signals during the reproductive cycle in vocal fish. Thus, sensory system plasticity allows animals to adapt to seasonal fluctuations in their environment and may be an evolutionarily important and conserved strategy for reproductive success and survival.

Although humans are not usually considered seasonal animals, seasonal changes in color perception and mood have been reported[25, 26]. Here we found significant gene expression changes in photopigments and the phototransduction cascade 1 week after transfer into summer-like conditions, potentially analogous to the reversal of depression found in seasonal affective disorder patients given light therapy[27]. Together, these data highlight the potential role of the retina and seasonal changes in photo-transduction in contributing to seasonal variation in behavior in humans and animals.

## Methods

**Ethics statement.** Animals were treated in accordance with the guidelines of the National Institutes of Natural Sciences, Japan. All experimental protocols were approved by the Animal Experiment Committee of the National Institutes of Natural Sciences, Japan.

**Animals.** Medaka fish (*O. latipes*) were obtained from a local dealer (Fuji 3A Project, Nagoya, Japan). For the behavioral experiments, medaka were kept under SC (short day and cool temperature; 10 h light/14 h dark and 8 °C) or LW (long day and warm temperature; 14 h light/10 h dark and 26 °C) conditions in a housing system (MEITO system, Meito Suien; LP-30LED-8CTAR, NK system) for at least 2 weeks. For the genome-wide transcriptome analysis, medaka were kept under SC conditions for 2 weeks and then transferred to LW conditions. *LWS*-null medaka were produced using the CRISPR/Cas9 system[21]. Sample size was established based on the standard in the field. No method was used to randomize animals between experimental groups. Neither investigators were blinded to each sample.

**Animal treatments.** Castration was performed following the protocol of Iwamatsu[28]. Medaka kept under SC conditions were deeply anesthetized with 0.05% 3-aminobenzoic acid ethyl ester methansulfonate salt. A 2–3 mm lateral incision was made along the abdomen with a razor and the testes were excised through it. At the end of the surgery, the incision was closed with surgical adhesive (Toagosei Co., Ltd). For sham-operated fish, the same operation except for removal of the testes was performed. After 2 weeks of recovery, fish were transferred into LW conditions. Fish were sacrificed 1 week after transfer into LW conditions and both eyes were collected for qPCR analyses. In the steroid hormone experiments, medaka kept under SC conditions were exposed to 100 µg l$^{-1}$ of T, E2 (Sigma-Aldrich) or MT (Wako Pure Chemical Industries, Ltd) dissolved in ethanol for 2 weeks. The concentration of ethanol did not exceed 0.1% of the total volume of water and the water containing ethanol or hormone was changed every 48 h. After 2 weeks of treatment, tissue samples were collected. This treatment has been shown to increase serum levels of the hormones[29]. In other experiments, medaka kept under SC conditions were transferred into SC or LW conditions for 1 week.

**Behavioral assays.** To record the behavior of SC or LW fish in a home tank, two fish (male and female) were habituated to a tank (15 cm square) for 1 day under SC or LW conditions. The fish were then introduced into a test tank of the same size and water temperature, and locomotor activity was recorded for 10 min using a 3D monitoring system (Dipp-AAM, Ditect). The 3D data were analyzed in R[30] (Fig. 1a). To examine phototaxis, male fish were taken directly from their home

tank and then placed into a test tank of the same water temperature (15 cm square), in which light was provided from one side (Fig. 1b). After habituation for 3 min, behavior was recorded for 1 min before and after lights out using the 3D monitoring system. To measure the attractiveness of orange–red-colored breeding female, we constructed a 3D-CG model fish by adding natural biological motions to a 3D female fish (Fig. 1c and Supplementary Movie 1)[31–33]. By eliminating color, we also generated a gray 3D-CG model fish with the same shape and motions (Supplementary Fig. 2c). As males normally approach females and this triggers reproductive behavior in medaka[34], we used males as test fish. Male fish were singly habituated in a tank (15 cm square) for 1 day under SC or LW conditions. 3D-CG fish started to move on the screen and the behavior of the test fish was monitored for 5 min. The time located on the screen side of the tank (within 1 cm distance of the wall) was measured using tracking software designed in-house (Medaka Fish Tracker ver. 3.7, available at http://www.nibb.ac.jp/neurophys/download/). The data collected 1 min before the appearance of the 3D-CG fish on the screen were used as a baseline.

**Microarray experiments**. Medaka microarrays (Custom Gene Expression Microarray 4 × 44 K, Agilent Technologies), which contain more than 31,000 probes, were used for these experiments. For each time point, both eyes were collected from six animals. Total RNA was prepared from three fish ($n = 2$) to duplicate our observations on two separate arrays using the RNeasy tissue kit (QIAGEN). Complementary DNA synthesis and cRNA labeling reactions were performed with the Low Input Quick Amp Labeling Kit (Agilent Technologies). Labeled cRNA was purified with RNeasy mini spin columns (QIAGEN) and hybridized using the Gene Expression Hybridization Kit (Agilent Technologies). After washing with the Gene Expression Wash Buffer (Agilent Technologies), the glass slide was scanned on a Microarray Scanner (Agilent Technologies).

**Quantitative PCR**. Reverse transcription was performed on total RNA (200 ng) using ReverTra Ace (Toyobo) and oligo-dT primers. Samples contained SYBR Premix Ex Taq II (Takara), 0.4 μM gene-specific primers (Supplementary Table 1) and 2 μl synthesized cDNA in 20 μl. qPCR was performed on Applied Biosystems 7500 Real-Time PCR System (Tokyo, Japan) as follows: 95 °C for 30 s, followed by 40 cycles of 95 °C for 5 s and 60 °C for 30 s. The *RPL7* gene was used as an internal RNA control.

**In situ hybridization**. Based on the previous studies[35], the eyes of SC and LW fish were fixed in 4% paraformaldehyde in phosphate-buffered saline (pH 7.4) and paraffin-embedded sections were cut at a thickness of 7 μm. Probe templates were amplified with *LWS*-specific primers (Forward: 5′-gacctgatgtgttcagtggaagc-3′; Reverse: 5′-cctctttgtcctcatttttggaaac-3′′) and cloned into pCR-II-TOPO (Invitrogen), followed by linearization and in vitro transcription (Roche Life Science). Sections were treated with proteinase K (4 μg ml⁻¹) and subsequently hybridized with DIG-labeled RNA probes. Hybridization signals were detected using alkaline phosphatase-conjugated anti-DIG antibody (Roche Life Science), with Nitro Blue tetrazolium and 5-bromo-4-chloro-3-indolyl-phosphate as chromogenic substrates.

**Measurement of spectral data**. We sacrificed adult fish using ice ($n = 4$) and placed in a Petri dish. The Petri dish was then placed on a ColorChecker 18% Gray Balance (X-rite). We used halogen lamp (Toshiba, JR12V50WF/K5FEZ) to produce incident light. The reflectance spectrum of real fish and the spectral power distribution of 3D-CG model fish on the screen were obtained using an SOC710-VP Hyperspectral Imager (Surface Optics) at wavelength of 400–800 nm on the caudal fin, where the nuptial coloration is most obvious. Relative radiant output of white light and monochromatic red light was obtained by HIDAMARI mini S-2440C (Soma Optics, LTD).

**Statistical analysis**. *F*-tests were used to determine variance. The data with a normal distribution were analyzed by a Student's *t*-test between two groups, whereas one-way ANOVA was used to compare three or more groups. Where variance was significantly different between groups, a Welch's *t*-test was used. All data were analyzed by the statistical software program Statcel2. Microarray data were analyzed using the GeneSpring software (Agilent Technologies).

**Code availability**. The computer code for 3D-CG medaka is available on Figshare at https://doi.org/10.6084/m9.figshare.4822243 and https://doi.org/10.6084/m9.figshare.4822234.

**Data availability**. The microarray data are available at NCBI Gene Expression Omnibus (accession number GSE94258). All other data are available from the authors upon request.

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

## Acknowledgements

We thank the NBRP-Medaka (National Bio-Resource Project of MEXT, Japan), the Data Integration and Analysis Facility, the Functional Genomics Facility, the Spectrography and Bioimaging Facility, and the Model Plant Research Facility of NIBB, Japan, for use of their facilities. We also thank M. Okubo, A. Akama, N. Baba, C. Kinoshita and M. Kondo for technical assistance, and Drs T.K. Tamai, Y. Nakane and T. Nishiwaki-Ohkawa for comments on the manuscript. This work was supported by JSPS KAKENHI Grant-in-Aid for Specially Promoted Research (26000013), the Human Frontier Science Program (RGP0030/2015) and NIBB Priority Collaborative Research Project (16-101). WPI-ITbM is supported by the World Premier International Research Center Initiative (WPI), MEXT, Japan.

## Author contributions

T.Y. conceived the research, and T.Y. and T. Shimmura designed the project. T. Shimmura performed all behavioral assays. T. Nishimura and M.T. provided medaka microarrays. T. Shimmura, T. Shimo, T. Senga and T.Y. performed the microarray analysis. T. Shimmura and T. Nakayama performed qPCR analysis. A.S. performed in situ hybridization. S.F. provided *LWS*-null fish, and T. Shimmura and T. Shimo genotyped *LWS*-null medaka. M.Y. and E.W. constructed behavioral assay system using 3D-CG fish. S.F., Y.K., E.W., M.Y. and T. Nakayama measured the spectral data. K.N. provided new materials. T.Y., T. Shimmura and T. Nakayama wrote the manuscript. All authors discussed the results and commented on the manuscript.

## Additional information

**Competing interests:** The authors declare no competing financial interests.

