## [Peer review file · Nature Communications]

Reviewer #1 (Remarks to the Author):

In their MS "Dynamic plasticity in the phototransduction regulates seasonal changes in color perception" Shimmura et al. use behavioural experiments and microarray data to show that Medaka exhibit seasonal variations in visual-guided behaviours which qualitatively match seasonal changes in retinal gene expressions. Using genetic silencing of the LWS opsin they go on to show that the animal's "red" opsin channel appear to be important in this switch.

Overall, there is a lot to like. The notion that sensory processes can vary not only between species but also within species depending on their sensory environments or season is an important aspect in our understanding of sensory ecology and how animals adapt to changes in their environment on physiological time scales. The use of Medaka's mate/light-attraction behaviour combined with studies of gene expression provide a nice angle to survey what might be going on at a genetic level.

The study is generally well designed and executed and the writing is, for the most part, of a high standard and clear. However, before considering this work for publication, there are a few points that ought to be addressed.

Major

1) What is the spectrum of the red/orange mating colouration, and how does this superimpose with the LWS (and other opsins)? This should be trivial to measure and would add a lot of substance to this story. Also, what is the spectrum of the "white" light used? Next, how does the spectrum of "red" of the virtual fish (presented on a computer monitor) match to the LWS opsin? The key reason all this matters is that spectra may vary widely between different light sources, and it is very possible that the LWS opsin(s) are not the ones primarily in range for the presented stimuli. Moreover, chromatic content is (almost) always detected by chromatic contrast (e.g. comparing "green" vs "red" opsin-driven channels), rather than "detected" (e.g. just using the red opsin). As such, it is critical to understand which opsins should matter in the author's paradigm to be able to interpret the gene expression changes as well as the results of the LWS silencing.

2) I would suspect that while opsin expression changes - or, more generally, "cone changes" - are important in this behavioural switch, they are not the only thing that's going on. I should like to see more mention of other processes, for example possible changes in retinal computations or central weighing of retinal inputs driven by changes in gene expression. For example, one key thing that seems to always matter in the retina in these kind of situations are gap junctions and dopamine regulation. Do the authors see retinal changes in Connexins or dopamine receptors? A careful expansion of the (existing) microarray results' presentation to test for these kind of changes would be welcomed. If something pops out, it may well be worth doing the relevant in-situ (or immunos) like in Fig. 2c.

Minor

3) Line 61, "mediate" (x2). Wrong term. "Underlie", "Are used/critical in"? Please rephrase

4) The authors appear to struggle with the correct English use of the word "the". For example, in the title, there should be no "the" in front of phototransduction. Conversely, several times throughout the text there are articles missing. For example in front of "LD stimulus" in lines 128, 133, 141 (x2) etc. This should be carefully checked.

5) I would advise against the use of the terms "color" and "perception" which are subjective terms used mostly in psychology. For this context, it would be more accurate to use "spectral content/composition; wavelength" and "detection" or synonyms

6) Line 177 - "costly". Yes, perhaps it is costly, but the authors have not shown this "cost". Please

clarify that this is conjecture.

7) Line 99,100 – for one, believe is probably the wrong term here. Speculate/suggest? Second, if the fish are “seeing the world in a different way” is not the point, if anything they “sample/process the visual environment differently” – seeing, again, is subjective.

8) Is the (very useful!) 3D-fish code open access? If not, I would strongly recommend providing this access.

Reviewer #2 (Remarks to the Author):

In this well-structured and clearly written study of visual ecology in Medaka (*Oryzias latipes*), the authors use a multidisciplinary approach to investigate the effect of seasonality on color perception and demonstrate the plastic nature of the visual system. Genome-wide expression analysis using microarrays highlighted seasonal changes in expression pattern of both photopigment (opsins) and the downstream phototransduction pathway genes. Based on the observation of Medaka fish in their natural environment showing a different performance between seasons, behavioral experiments revealed that only light environments simulating summer mediated phototaxis. Moreover, also the attractiveness to orange-red-colored 3D-computer-modelled fish representing the natural nuptial coloration during breeding (summer) season, is increased in light environments simulating summer. Interestingly, CRISPR/Cas9 modulated fish lacking the LWS opsin (long wavelength-sensitive opsin gene) resulted in a deviant light response behavior and a weaker preference for orange-red-colored model fish when compared to the wild-type fish.

This study is a significant contribution to field of visual molecular ecology but also to the wider field, given its diverse methodical scope (transcriptome analysis, behavioral assay combined with the CRISPR/Cas9 system, gene expression, in situ hybridization), and thus its ability to link the molecular machinery to a natural behavior. It clearly demonstrates that vision is a complex process, and that in order to unravel its complexity, it needs a multidisciplinary approach – an attitude that is also eligible in other research disciplines. Hence, this is an interesting study of dynamic plasticity in the visual perception and visual ecology, and I only have two major and a few minor points to be clarified and addressed. In regards to the appropriateness and validity of statistical analysis used I do not have concerns. However, I recommend to provide more detail on the in situ hybridization pictures (see major points 2)), as well as on spectral composition of the modeled fish used (see minor points 7)).

Major points:

1) The in situ hybridization pictures of Fig.2C show that LWS is present in the photopigment layer in only LD fish. However, it has been shown that opsins present in double and single cone members can be spatially variable throughout the retina (see Dalton et al., 2014; 2015; 2016), and this variation has even been reported to be influenced by the light environment fish are raised (Dalton et al., 2015 – I highly suggest to cite this paper as it is showing that the spectral sensitivity - including LWS expression - is affected by environmental background spectra potentially influencing behaviors such as courtship and speciation). It would therefore be highly interesting whether the expression of LWS is uniformly present throughout LD retinas respectively uniformly absent in SL retinas? Or in other words, do the pictures presented in Fig.2C show the same region of the retina (a fact that would make the result even more powerful)? Please indicate in the legend which part (dorsal, ventral, central?) of the retina is illustrated, and whether the same retinal part for both LD and SL fish is presented.

2) Moreover, all visual opsins showing expression differences between LD and SL treatments, and are presented in the text respectively in Extended Data Fig.2 and Extended Data Fig.3 are up-

regulated in LD fish. This suggests that less opsins are present in the photoreceptor cells during treatments simulating winter. Together with the in situ hybridization results highlighting that LWS is induced only in LD fish, I wonder whether photoreceptor cells in SL fish are empty and thus non-functional. Can you comment on whether non-labeled photoreceptor cells in SL fish (Fig. 2C) are actually "empty" or do they occupy another opsin type? Also, can you present higher resolution pictures or comment on whether one or both double cone members express LWS in SL and LD fish? I wonder whether the higher presence of LWS in LD fish is caused from LWS being expressed in only one double cone member in SL fish and in both double cone members in LD fish? In this case one might expect to find another opsin type being expressed in one member of double cones in SL fish. However, this would imply that at least one opsin should be down-regulated in LD fish.

Minor points:

1) Abstract: The observation of fish in the wild is that compared to summer, winter fish are less active and stay at the riverbed (line 175/176). Only the artificial created SL and LD tanks simulating winter respectively summer show the behavioral outcomes. In this context, the two statements that "Medaka were active and exhibited clear phototaxis in summer, but remained at the bottom of the tank and failed to exhibit phototaxis in winter." and "[...] that Medaka are more attracted to orange-red-colored model fish in summer than in winter." should be rephrased (e.g. replace "summer" with "light/water environment simulating summer")

2) Line 60: Rhodopsin in rods mediates [...]

3) Line 82: I was unclear to me, until I have read the methods, what characteristics (light and temperature conditions) the novel tank had. Thus, it would be helpful for the reader that a note (e.g. "see methods") is added.

4) Fig 1B Top: The upper part is showing examples of the LD fish, isn't it? If yes, I suggest to mention that in the figure legend: e.g. "Medaka (LD fish) exhibiting [...]".

5) Extended Data Fig. 3:

- I guess the legend of the Y-Axis can be redone the way that the relative mRNA level is related to both up- and downregulated genes.

- To my understanding the figure shows fish on day 0 from SL conditions and fish 14 days after being transferred from SL to LD conditions. The way the figure legend is written, it might be a bit unclear to the reader that only LD fish have been transferred. My suggestion would be to reorganize the sentence to "[...] using Medaka on days 0 (SL) and 14 after transfer from SL to LD (LD)." Perhaps you can even name the transferred fish tLD in the figure itself?

6) Line 157: It would be good for the reader to mention the spectra (λ_{max}) of the white light stimulus that is used as a control to clarify that the white light is also composed of the long wavelength spectra as used for the red light stimulus (at least I hardly assume it is).

7) Is the red-orange coloration of the real Medaka fish (Line 87) been measured using spectrophotometry? Accordingly, is the spectral reflectance of the orange-red-colored modeled fish (e.g. Line 98) reflecting the wild-type coloration? And last, what monochromatic fish models (Line 96; Fig. 1D; Fig.3B) have been used? I find it essential to provide those information at least in the methods.

8) Line 176/177: The authors suggest that during winter months, Medaka is less active to save energy related to visual tasks. Out of curiosity, what about other visually mediated behaviors like feeding behavior during the winter months? Is that also reduced? Again, this question is related to the point above as all opsins seem to be down-regulated in winter.

Reviewer #3 (Remarks to the Author):

Motivated in part by recent findings that human color vision changes across the seasons, the authors examine seasonal changes in photoreceptor function in Medaka fish. They simulate both summer and winter environments by altering both day/night cycle length and temperature (LD vs SD). They report several findings including:

1: Behavior changes in LD vs SD fish: specifically, behavior related to mate seeking and light approach / avoidance.

2: Gene expression is altered: many genes, including those relating to both photoreceptor opsin expression and post-receptor processing have altered expression pattern in the two conditions.

They then perform a knockout manipulation to ask whether one of the LWS photoreceptors in the fish eye is important behaviorally. They find that photoreceptor knockout fish have altered behavioural patterns including mate seeking.

These are an interesting set of experiments but I am not sure that they form a coherent 'whole' that demonstrates the point that the authors are trying to prove.

The most obvious issue is that both light duration and water temperature are altered between the summer and winter conditions. Although the authors measure behavioral and widespread transcriptional effects, it is not clear what causes them. There might even be subtle effects going on here like the fact that dissolved oxygen levels might be quite a bit higher in colder water.

So the authors have shown that something alters both gene expression and behavior between simulated summer and winter conditions. For example, fish in the 'winter' condition (SD) tend to sit at the bottom of the tank (Fig 1a). But this could be due to all sorts of reasons – perhaps their vision is identical in the winter but they are simply much more sluggish? A related effect is that SD fish tend not to exhibit light-directed motion (spending, on average, the same amount of time in the lit and unlit sides of the tank) and also do not change this behavior after 'lights out' (1B). But 1a strongly suggest that this might be because they just do not move much at all.

The observation that 'LD' fish tend to swim towards 'orange' fish simulations is also interesting but, again, may be confounded with the general lack of motion seen in SD fish. It is also not quite clear that the 3D CRT models are good simulations of real fish. These animals have many photoreceptors (EDFig 4) while CRTs have just three primaries. While the replicants and 'real' orange fish look similar to humans (because we also have 3 photoreceptors) they are unlikely to be realistic copies for an actual fish. Colored CRT fish somehow appear different to achromatic fish but the fact that the authors do not attempt to, for example, compute the photoreceptor catches in the multiple different photoreceptors suggests that they have not thought about why this might be. They don't even provide the CRT spectra so no one else can make these estimates either. I can't help thinking that to do this color experiment effectively, you might need to sacrifice spatial resolution and use a multi-primary light source that can generate spectra that isolate responses in particular fish cone classes.

Figure 2 shows that the transcription of over 1500 genes alters between summer and winter and that some of these changes occur in genes involved in visual transduction (including the LWS opsin). LWS opsin, for example, is up-regulated significantly in the LD fish. In Figure 3, the authors show the behaviours of LWS +/+ and LWS knockouts in different lighting conditions.

In Figure 1, we learned that LD fish exhibit negative phototaxis, swimming away from a white light. White light presumably contains power at many different wavelengths. In F3 we learn that LWS knockouts tend to swim away from white light a little less and actually swim towards a reddish light. We are not provided with the amplitude spectra of the 'white' and 'red' lights (the 'red' is presumably an LED with a Gaussian response centered on 730nm) so again it is hard to pin down exactly what is going on here. It is possible that the LWS opsin is almost exclusively responsible for negative phototaxis but in that case why do LWS $-/-$ fish also stay away from white light (which presumably has plenty of long wavelengths in it)? Alternatively, LWS opsins might contribute only a little bit towards phototaxis (as you might conclude from 3A, left panel) but then why does removing the short wavelengths lead to almost no change in behavior in the LWS $+/+$ animals? Figure 3b shows that that LWS $+/+$ fish tend to swim towards orange CRT simulations more than 'achromatic' simulations and here the effect seems to be clearer so presumably the orange CRT spectra do preferentially stimulate the LWS opsin but it's noticeable that LWS $-/-$ fish also seem to prefer the orange simulations. I can't quite tell what the * refers to in this panel. In 3a it seems to indicate a within-light-condition change across genotypes. In 3b surely the most informative test would be a within-genotype change across light conditions? It does sort of look like LWS $-/-$ fish also prefer orange mates but I can't tell if this is significant.

In short, I am convinced that some combination of light exposure and water temperature alters gene expression in these fish. These things (along with others – for example, the light spectrum) do change across seasons. I also believe that there are behavioral differences associated with these gene expression changes. And that a small subset of the genes whose expression is changed are involved in phototransduction. Beyond that, the link between color vision and behavior is not controlled well enough to yield any insights. The stimulus display system and lighting changes are not well-characterized and is almost certainly unable to simulate the real colors of fish in the wild. And the behavioral choice differences seem to be conflated with huge differences in overall activity making comparisons between LD and SD conditions very difficult. I think spectral measurements of both the broad-field white illuminations and the CRT stimuli, combined with information about the individual photoreceptor absorption curves might go some way towards contextualizing the observed behavior but it is going to be difficult to dissociate the temperature and light-dependent changes in gene expression. Finally, the link to human SAD is, I think, fairly tenuous and could be removed without impacting the science.

Reviewer #4 (Remarks to the Author):

Seasonal changes are truly ubiquitous and cover virtually every aspect of life on this planet albeit 99% of the published work relates solely to reproduction. This fascinating paper from Yoshimura's group extends our knowledge into photoperiodic changes in color perception. As with all his work this contribution genuinely represents modern biological research "at the cutting edge": I note he is now using CRISP-R technology.

He has developed a new model system using Medaka fish that change their behavior from summer to winter and which can be reproduced experimentally by altering photoperiod. From the account offered it appears robust and quantifiable. Having established this platform he has then shifted fish to long days and at various times thereafter (days) he has analyzed retinal alterations in the families of opsin genes and in some of the downstream pathways. Many genes are altered (as one would expect) and he has focused for this paper on the so-called LWS genes. By using CRISP-R/CAS technologies he has altered expression of the LWS genes and produced statistically significant alterations in behavior.

I should have been proud to have offered this paper. Most importantly it is fascinating and interesting to a wide readership - what Nature is all about!. Secondly it mixes a range of techniques and approaches which represent the 21st Century.

PUBLISH IT!

Point-by-point responses to reviewers' comments to Shimmura *et al.*, “Dynamic plasticity in phototransduction regulates seasonal changes in color perception”

We are grateful to all of the reviewers for their valuable comments, which helped us to significantly improve our manuscript.

Below, the reviewers' comments are shown in blue, and our answers to their comments are shown in black. As requested by Reviewer 3, we have performed additional experiments and discovered that temperature, not daylength, is upregulating *LWS* gene expression. We therefore changed throughout the text, LD to LW conditions, referring to long day/warm temperature conditions. We also renamed “monochromatic model fish” to “gray model fish” for accuracy.

We have also reformatted our manuscript to comply with the *Nature Communications* format.

Reviewer #1

In their MS “Dynamic plasticity in the phototransduction regulates seasonal changes in color perception” Shimmura *et al.* use behavioural experiments and microarray data to show that Medaka exhibit seasonal variations in visual-guided behaviours which qualitatively match seasonal changes in retinal gene expressions. Using genetic silencing of the *LWS* opsin they go on to show that the animal's “red” opsin channel appear to be important in this switch.

Overall, there is a lot to like. The notion that sensory processes can vary not only between species but also within species depending on their sensory environments or season is an important aspect in our understanding of sensory ecology and how animals adapt to changes in their environment on physiological time scales. The use of Medaka's mate/light-attraction behaviour combined with studies of gene expression provide a nice angle to survey what might be going on at a genetic level.

The study is generally well designed and executed and the writing is, for the most part, of a high standard and clear. However, before considering this work for publication, there are a few points that ought to be addressed.

Major

1) What is the spectrum of the red/orange mating colouration, and how does this superimpose with the *LWS* (and other opsins)? This should be trivial to measure and would add a lot of substance to this story. Also, what is the spectrum of the “white” light used? Next, how does the spectrum of “red” of the virtual fish (presented on a computer monitor) match to the *LWS* opsin? The key reason all this matters is that spectra may vary widely between different light sources, and it is very possible that the *LWS* opsin(s) are not the ones primarily in range for the

presented stimuli. Moreover, chromatic content is (almost) always detected by chromatic contrast (e.g. comparing “green” vs “red” opsin-driven channels), rather than “detected” (e.g. just using the red opsin). As such, it is critical to understand which opsins should matter in the author’s paradigm to be able to interpret the gene expression changes as well as the results of the LWS silencing.

Response: We have provided spectral data, as suggested by the reviewer, including relative radiant output of the light sources (Supplementary Fig. 1a), spectral reflectance of orange-red coloration of real fish (Supplementary Fig. 1b), spectral power of virtual fish from the monitor (Supplementary Fig. 1c), and absorption spectra of various opsins (Supplementary Fig. 1d).

We believe that the LWS opsins are playing a crucial role in recognizing the orange-red hues of real and virtual fish for the following reasons. The absorption maxima of the LWS opsins is 560 nm. White light, the orange-red color of real fish, and the orange-red of 3D-CG fish contain significant amounts of light at $\lambda \geq 560$ nm, so there is sufficient overlap with the LWS opsins (Supplementary Fig. 1). Furthermore, there is a significant reduction in shorter wavelengths of light in both real and virtual orange-red colored fish, and this should be recognized as a chromatic contrast between LWS and other cone opsins. Together, this indicates that LWS is the photopigment that plays an important role in recognizing orange-red hues. (Note: It is technically quite difficult (and maybe impossible) to perfectly mimic the orange-red color of real fish with the orange-red of 3D-CG fish. There are certainly limitations to using computer graphics and computer-generated images for behavioral studies, but we have taken all this into consideration and optimized our assay. In our experiments, what we were able to compare was the effect of red light from the monitor by changing the ratio of red light in the 3D-CG fish, and we are confident that our results are solid and convincing.)

Although the wavelength of red light we used ($\lambda = 730$ nm) is much longer than the λ_{max} of LWS (560 nm), wild-type Medaka have been shown to respond behaviorally to light at unexpectedly long wavelengths (up to 830 nm) under light-adapted conditions (Homma et al., 2017). This study also demonstrated that *LWS*-null Medaka could respond only up to 740 nm, demonstrating that no other cone opsins except LWS could be responsible for absorbing light at wavelengths greater than 740 nm. Monochromatic red light at 730 nm should be readily detected by the wild-type Medaka, whereas *LWS*-null Medaka are much less sensitive to this wavelength and should have difficulty detecting this light. We believe that the response of the green (and other) cone opsins to this red light is too weak to induce negative phototaxis. Therefore, the red light-induced negative phototaxis we observe in wild-type Medaka should depend solely on the LWS cones.

We have added this information in the Results section.

2) I would suspect that while opsin expression changes - or, more generally, “cone changes” - are important in this behavioural switch, they are not the only thing that’s going on. I should like to see more mention of other processes, for example possible changes in retinal computations or central weighing of retinal inputs driven by changes in gene expression. For example, one key thing that seems to always matter in the retina in these kind of situations are gap junctions and dopamine regulation. Do the authors see retinal changes in Connexins or dopamine receptors? A careful expansion of the (existing) microarray results’ presentation to test for these kind of changes would be welcomed. If something pops out, it may well be worth doing the relevant in-situ (or immunos) like in Fig. 2c.

Response: Unfortunately, we did not observe any significant changes in genes involved in retinal coupling or processing, such as connexins (CX36, CX45 and CX57) or dopamine receptors.

Minor

3) Line 61, “mediate” (x2). Wrong term. “Underlie”, “Are used/critical in”? Please rephrase

Response: Rewritten as suggested.

4) The authors appear to struggle with the correct English use of the word “the”. For example, in the title, there should be no “the” in front of phototransduction. Conversely, several times throughout the text there are articles missing. For example in front of “LD stimulus” in lines 128, 133, 141 (x2) etc. This should be carefully checked.

Response: Carefully corrected by a native English speaker.

5) I would advise against the use of the terms “color” and “perception” which are subjective terms used mostly in psychology. For this context, it would be more accurate to use “spectral content/composition; wavelength” and “detection” or synonyms

Response: We appreciate the reviewer's comments on the use of the term "color perception." Although the alternatives suggested by the reviewer might be more detailed or specific, they are not commonly used. We believe "color perception" would attract a wider audience, and we would prefer to use this term, although we will leave the final decision up to the editor.

6) Line 177 – “costly”. Yes, perhaps it is costly, but the authors have not shown this “cost”. Please clarify that this is conjecture.

Response: Because Medaka rarely eat in winter, we believe they need to conserve energy. We have added this information in Discussion as follows: “Because Medaka rarely eat in winter, we predict that they need to save energy. We speculate from our results that Medaka could be

saving energetically costly visual functions during winter by downregulating the expression of genes involved in phototransduction.”

7) Line 99,100 – for one, believe is probably the wrong term here. Speculate/suggest? Second, if the fish are “seeing the world in a different way” is not the point, if anything they “sample/process the visual environment differently” – seeing, again, is subjective.

Response: Corrected as suggested.

8) Is the (very useful!) 3D-fish code open access? If not, I would strongly recommend providing this access.

Response: Yes, it is. We have included a movie of a 3D-CG fish in the Supplementary section (Supplemental Movie 1) and have provided the computer code in the “Data availability” section.

Reviewer #2

In this well-structured and clearly written study of visual ecology in Medaka (*Oryzias latipes*), the authors use a multidisciplinary approach to investigate the effect of seasonality on color perception and demonstrate the plastic nature of the visual system. Genome-wide expression analysis using microarrays highlighted seasonal changes in expression pattern of both photopigment (opsins) and the downstream phototransduction pathway genes. Based on the observation of Medaka fish in their natural environment showing a different performance between seasons, behavioral experiments revealed that only light environments simulating summer mediated phototaxis. Moreover, also the attractiveness to orange-red-colored 3D-computer-modelled fish representing the natural nuptial coloration during breeding (summer) season, is increased in light environments simulating summer. Interestingly, CRISPR/Cas9 modulated fish lacking the LWS opsin (long wavelength-sensitive opsin gene) resulted in a deviant light response behavior and a weaker preference for orange-red-colored model fish when compared to the wild-type fish.

This study is a significant contribution to field of visual molecular ecology but also to the wider field, given its diverse methodical scope (transcriptome analysis, behavioral assay combined with the CRISPR/Cas9 system, gene expression, in situ hybridization), and thus its ability to link the molecular machinery to a natural behavior. It clearly demonstrates that vision is a complex process, and that in order to unravel its complexity, it needs a multidisciplinary approach – an attitude that is also eligible in other research disciplines. Hence, this is an interesting study of dynamic plasticity in the visual perception and visual ecology, and I only have two major and a few minor points to be clarified and addressed. In regards to the

appropriateness and validity of statistical analysis used I do not have concerns. However, I recommend to provide more detail on the in situ hybridization pictures (see major points 2)), as well as on spectral composition of the modeled fish used (see minor points 7)).

Major points:

1) The in situ hybridization pictures of Fig.2C show that LWS is present in the photopigment layer in only LD fish. However, it has been shown that opsins present in double and single cone members can be spatially variable throughout the retina (see Dalton et al., 2014; 2015; 2016), and this variation has even been reported to be influenced by the light environment fish are raised (Dalton et al., 2015 – I highly suggest to cite this paper as it is showing that the spectral sensitivity - including LWS expression - is affected by environmental background spectra potentially influencing behaviors such as courtship and speciation). It would therefore be highly interesting whether the expression of LWS is uniformly present throughout LD retinas respectively uniformly absent in SL retinas? Or in other words, do the pictures presented in Fig.2C show the same region of the retina (a fact that would make the result even more powerful)? Please indicate in the legend which part (dorsal, ventral, central?) of the retina is illustrated, and whether the same retinal part for both LD and SL fish is presented.

Response:

We have cited Dalton et al., 2015 as suggested.

Upregulation of *LWS* by LW conditions was observed in the whole retina. The images in Figure 2c depict expression in the central area of the retina. We have added the following sentence to the legend: “Representative images from the central area of the retina (Also see Supplementary Fig. 6)”. We have also provided higher magnification images of the dorsal, central, and ventral retina in the Supplementary Figure 6 as requested.

2) Moreover, all visual opsins showing expression differences between LD and SL treatments, and are presented in the text respectively in Extended Data Fig.2 and Extended Data Fig.3 are up-regulated in LD fish. This suggests that less opsins are present in the photoreceptor cells during treatments simulating winter. Together with the in situ hybridization results highlighting that LWS is induced only in LD fish, I wonder whether photoreceptor cells in SL fish are empty and thus non-functional. Can you comment on whether non-labeled photoreceptor cells in SL fish (Fig. 2C) are actually “empty” or do they occupy another opsin type? Also, can you present higher resolution pictures or comment on whether one or both double cone members express LWS in SL and LD fish? I wonder whether the higher presence of LWS in LD fish is caused from LWS being expressed in only one double cone member in SL fish and in both double cone members in LD fish? In this case one might expect to find another opsin type being expressed in one member of double cones in SL fish. However, this would imply that at least one opsin

should be down-regulated in LD fish.

Response:

Because expression of most of the image forming-visual pigments is significantly downregulated under SL conditions (Figure 2, Supplementary Figure 4, 5), we speculate that most of the photoreceptors are nearly “empty” in winter.

The reviewer makes an interesting point about which cone cell-types are expressing *LWS*. We have provided higher resolution pictures of the *in situ* hybridization using the Differential Interference Contrast (DIC) microscope in Supplementary Figure 6. However, it was difficult to distinguish the cell types that express *LWS* from these pictures. Although this is an interesting point, we believe that the lack of this information does not affect our conclusion.

Minor points:

1) Abstract: The observation of fish in the wild is that compared to summer, winter fish are less active and stay at the riverbed (line 175/176). Only the artificial created SL and LD tanks simulating winter respectively summer show the behavioral outcomes. In this context, the two statements that “Medaka were active and exhibited clear phototaxis in summer, but remained at the bottom of the tank and failed to exhibit phototaxis in winter.” and “[...] that Medaka are more attracted to orange-red-colored model fish in summer than in winter.” should be rephrased (e.g. replace “summer” with “light/water environment simulating summer”)

Response: We have rephrased to “conditions simulating summer”.

2) Line 60: Rhodopsin in rods mediates [...]

Response: “mediates” has been changed to “underlies” as suggested by Reviewer #1.

3) Line 82: I was unclear to me, until I have read the methods, what characteristics (light and temperature conditions) the novel tank had. Thus, it would be helpful for the reader that a note (e.g. “see methods”) is added.

Response: “see Methods section” has been added as suggested.

4) Fig 1B Top: The upper part is showing examples of the LD fish, isn't it? If yes, I suggest to mention that in the figure legend: e.g. “Medaka (LD fish) exhibiting [...]”.

Response: Corrected to “Medaka (LW fish) exhibiting” as suggested.

5) Extended Data Fig. 3:

- I guess the legend of the Y-Axis can be redone the way that the relative mRNA level is related to both up- and downregulated genes.

Response: “relative mRNA level” was rewritten as suggested.

- To my understanding the figure shows fish on day 0 from SL conditions and fish 14 days after being transferred from SL to LD conditions. The way the figure legend is written, it might be a bit unclear to the reader that only LD fish have been transferred. My suggestion would be to reorganize the sentence to “[...] using Medaka on days 0 (SL) and 14 after transfer from SL to LD (LD).” Perhaps you can even name the transferred fish tLD in the figure itself?

Response: The figure legend was rephrased as suggested.

6) Line 157: It would be good for the reader to mention the spectra (λ_{max}) of the white light stimulus that is used as a control to clarify that the white light is also composed of the long wavelength spectra as used for the red light stimulus (at least I hardly assume it is).

Response: We have provided the relative radiant output of the light sources in Supplementary Fig. 1a as requested.

7) Is the red-orange coloration of the real Medaka fish (Line 87) been measured using spectrophotometry? Accordingly, is the spectral reflectance of the orange-red-colored modeled fish (e.g. Line 98) reflecting the wild-type coloration? And last, what monochromatic fish models (Line 96; Fig. 1D; Fig.3B) have been used? I find it essential to provide those information at least in the methods.

Response:

As shown in Supplementary Fig. 1b, we have provided spectral reflectance data of the orange-red coloration of real fish. We have also provided spectral power distribution of orange-red-colored model fish and gray model fish in Supplementary Fig. 1c. Real fish as well as orange-red-colored model fish had high spectral power at long wavelengths covered by LWS. Note that gray model fish also contained spectral power in the red range (Supplementary Fig. 1c). However, this is because gray is produced by mixing three-additive primary color beams; red, green and blue (RGB). We have added an explanation of this in the legend of Supplementary Fig. 1.

Also note: It is technically quite difficult (and maybe impossible) to perfectly mimic the orange-red color of real fish with the orange-red of 3D-CG fish. There are certainly limitations to using computer graphics and computer-generated images for behavioral studies, but we have taken all this into consideration and optimized our assay. In our experiments, what we were able to compare was the effect of red light from the monitor by changing the ratio of red light in the 3D-CG fish, and we are confident that our results are solid and convincing.

8) Line 176/177: The authors suggest that during winter months, Medaka is less active to save

energy related to visual tasks. Out of curiosity, what about other visually mediated behaviors like feeding behavior during the winter months? Is that also reduced? Again, this question is related to the point above as all opsins seem to be down-regulated in winter.

Response: Although Medaka have the capacity to move in winter (See Supplementary Fig. 2), they rarely eat during this period. It is not yet clear whether reduced food intake is caused by changes in visual function. However, it seems likely that they do need to save energy. Therefore, we have added this information to the Discussion as follows: “Because Medaka rarely eat in winter, we predict that they need to save energy. We speculate from our results that Medaka could be saving energetically costly visual functions during winter by downregulating the expression of genes involved in phototransduction.”

Reviewer #3

Motivated in part by recent findings that human color vision changes across the seasons, the authors examine seasonal changes in photoreceptor function in Medaka fish. They simulate both summer and winter environments by altering both day/night cycle length and temperature (LD vs SD). They report several findings including:

1: Behavior changes in LD vs SD fish: specifically, behavior related to mate seeking and light approach / avoidance.

2: Gene expression is altered: many genes, including those relating to both photoreceptor opsin expression and post-receptor processing have altered expression pattern in the two conditions. They then perform a knockout manipulation to ask whether one of the LWS photoreceptors in the fish eye is important behaviorally. They find that photoreceptor knockout fish have altered behavioural patterns including mate seeking.

These are an interesting set of experiments but I am not sure that they form a coherent ‘whole’ that demonstrates the point that the authors are trying to prove.

The most obvious issue is that both light duration and water temperature are altered between the summer and winter conditions. Although the authors measure behavioral and widespread transcriptional effects, it is not clear what causes them. There might even be subtle effects going on here like the fact that dissolved oxygen levels might be quite a bit higher in colder water.

Response: We have performed additional experiments to determine the factors regulating *LWS* expression and found that neither photoperiod nor gonadal steroids have a significant effect. It is, in fact, temperature that is the primary factor upregulating *LWS* in summer (See Results section and Figure 4).

So the authors have shown that something alters both gene expression and behavior between

simulated summer and winter conditions. For example, fish in the ‘winter’ condition (SD) tend to sit at the bottom of the tank (Fig 1a). But this could be due to all sorts of reasons – perhaps their vision is identical in the winter but they are simply much more sluggish? A related effect is that SD fish tend not to exhibit light-directed motion (spending, on average, the same amount of time in the lit and unlit sides of the tank) and also do not change this behavior after ‘lights out’ (1B). But 1a strongly suggest that this might be because they just do not move much at all.

Response: Although overall activity is indeed decreased under winter conditions in the home tank, these fish certainly have the capacity to move when transferred to new environmental conditions such as the test tank. We have therefore provided data for the distance and speed of movement in the novel test tank (See Supplementary Figure 2). The results revealed no significant difference in the movement between winter and summer in the test tank, demonstrating the ability of SL fish to move. Therefore, we are confident of our results that show seasonal changes in spectral detection.

The observation that ‘LD’ fish tend to swim towards ‘orange’ fish simulations is also interesting but, again, may be confounded with the general lack of motion seen in SD fish. It is also not quite clear that the 3D CRT models are good simulations of real fish. These animals have many photoreceptors (EDFig 4) while CRTs have just three primaries. While the replicants and ‘real’ orange fish look similar to humans (because we also have 3 photoreceptors) they are unlikely to be realistic copies for an actual fish. Colored CRT fish somehow appear different to achromatic fish but the fact that the authors do not attempt to, for example, compute the photoreceptor catches in the multiple different photoreceptors suggests that they have not thought about why this might be. They don’t even provide the CRT spectra so no one else can make these estimates either. I can’t help thinking that to do this color experiment effectively, you might need to sacrifice spatial resolution and use a multi-primary light source that can generate spectra that isolate responses in particular fish cone classes.

Response: As written above, SL fish have the ability to move.

We have also provided the spectra data for real fish and the 3D-CG medaka model in Supplemental Fig. 1.

It is technically quite difficult (and maybe impossible) to perfectly mimic the orange-red of real fish with the orange-red of 3D-CG fish. There are certainly limitations to using computer graphics and computer-generated images for behavioral studies, but we have taken all this into consideration and optimized our assay.

Although a multi-primary light source has been proposed for overcoming the limitations in chromaticity and spectral characteristics of RGB-based image reproduction systems, our orange-red colored model fish do contain a red color component that is covered primarily by

LWS (See Supplementary Fig. 1). The results demonstrated that the LW but not SL Medaka preferred the 3D-CG with more red light, demonstrating that an increased ratio of red light is preferable to LW Medaka, but not SL Medaka. As this reviewer correctly pointed out, we would need to sacrifice spatial resolution when a multi-primary light source is used. For example, an increase in the number of primary colors causes a decrease in light energy of each primary color. To reduce noise of each primary color, the number of primaries should be lower. In addition, low-frame-rate image displays of multi-primary light source induces temporal artifacts, such as visible flickers and color breakups. In a previous study, we found that smooth-moving biological motion (60 frames per second (FPS)) is a critical factor to attract Medaka fish compared to jerky types of biological motion (e.g., 15, 10, 5, and 1 FPS) (Nakayasu and Watanabe 2014). Taking all these factors into account, we are confident that we performed our experiments under the best possible conditions.

Figure 2 shows that the transcription of over 1500 genes alters between summer and winter and that some of these changes occur in genes involved in visual transduction (including the LWS opsin). LWS opsin, for example, is up-regulated significantly in the LD fish. In Figure 3, the authors show the behaviours of LWS *+/+* and LWS knockouts in different lighting conditions.

In Figure 1, we learned that LD fish exhibit negative phototaxis, swimming away from a white light. White light presumably contains power at many different wavelengths. In F3 we learn that LWS knockouts tend to swim away from white light a little less and actually swim towards a reddish light. We are not provided with the amplitude spectra of the ‘white’ and ‘red’ lights (the ‘red’ is presumably an LED with a Gaussian response centered on 730nm) so again it is hard to pin down exactly what is going on here. It is possible that the LWS opsin is almost exclusively responsible for negative phototaxis but in that case why do LWS *-/-* fish also stay away from white light (which presumably has plenty of long wavelengths in it)? Alternatively, LWS opsins might contribute only a little bit towards phototaxis (as you might conclude from 3A, left panel) but then why does removing the short wavelengths lead to almost no change in behavior in the LWS *+/+* animals?

Response: We have provided spectral data for “white” and “red” light. As shown in Supplementary Fig. 1a, white light contained short-wavelength light as well as long-wavelength light. Although the λ_{\max} of LWS is approximately 560 nm, LWS has been demonstrated to cover an unexpectedly wide range of wavelengths (up to 830 nm) under light-adapted conditions (Homma et al., 2017). We therefore used 730 nm monochromatic red light as a light source to minimize the effect of other opsins.

We believe that multiple photopigments are involved in the negative phototaxis. Attenuated phototaxis in LWS^{-/-} to white light (shown in Fig. 3a left panel) indicates partial involvement of

LWS in the white-light induced negative phototaxis. As this reviewer points out, the magnitude of negative phototaxis to white light and monochromatic red light were similar in LWS^{+/+} Medaka. The light intensity of white light and red light are 23.1 $\mu\text{mol}/\text{m}^2/\text{sec}$ and 13.3 $\mu\text{mol}/\text{m}^2/\text{sec}$, respectively. We consider that these light intensities are bright enough (sufficient) to induce a full response in LWS^{+/+} Medaka.

We have added an alternative explanation (underlined) in the Results as follows:

“Wild-type fish exhibited negative phototaxis in response to both white and red light (Fig. 3a). Since white light contains long wavelengths light (Supplementary Fig. 1a), LWS-null fish exhibited less negative phototaxis in response to white light (Fig. 3a left), suggesting reduced photosensitivity in LWS-null fish. Critically, LWS-null fish exhibited darkness-induced light-seeking behavior following the switch to monochromatic red light (Fig. 3a right), demonstrating that LW induction of LWS is required for negative phototaxis to red light”.

Figure 3b shows that LWS ^{+/+} fish tend to swim towards orange CRT simulations more than ‘achromatic’ simulations and here the effect seems to be clearer so presumably the orange CRT spectra do preferentially stimulate the LWS opsin but it’s noticeable that LWS ^{-/-} fish also seem to prefer the orange simulations. I can’t quite tell what the * refers to in this panel. In 3a it seems to indicate a within-light-condition change across genotypes. In 3b surely the most informative test would be a within-genotype change across light conditions? It does sort of look like LWS^{-/-} fish also prefer orange mates but I can’t tell if this is significant.

Response: Although LWS^{-/-} fish showed a tendency to prefer orange-red colored compared to gray fish in Fig. 3b, there was no significant difference ($P = 0.209$, *t*-test).

It has been demonstrated that color cue is perceived as the relative balance of brightness at multiple wavelengths rather than the brightness at a specific wavelength in the mate discrimination in Medaka (Utagawa et al., 2016). Indeed, body color of summer Medaka consists of various wavelengths of light, including orange and red (See Supplemental Figure 1b). Although multiple photopigments are required to perceive “color”, our results clearly demonstrated that LWS is indeed necessary to discriminate orange-red body coloration. We have added this information (underlined) in the Results section as follows:

“although LWS-null fish are fertile under LW conditions, their behavior was clearly affected, as they exhibited a weaker preference for orange-red-colored model fish than wild-type fish (Fig. 3b right). Furthermore, there was no difference in the preference of LWS-null fish to either orange-red-colored or gray model fish ($P = 0.21$, *t*-test). Thus, our results demonstrated that LW induction of LWS contributes to LW-induced mate preference.”

In short, I am convinced that some combination of light exposure and water temperature alters

gene expression in these fish. These things (along with others – for example, the light spectrum) do change across seasons. I also believe that there are behavioral differences associated with these gene expression changes. And that a small subset of the genes whose expression is changed are involved in phototransduction. Beyond that, the link between color vision and behavior is not controlled well enough to yield any insights. The stimulus display system and lighting changes are not well-characterized and is almost certainly unable to simulate the real colors of fish in the wild. And the behavioral choice differences seem to be conflated with huge differences in overall activity making comparisons between LD and SD conditions very difficult. I think spectral measurements of both the broad-field white illuminations and the CRT stimuli, combined with information about the individual photoreceptor absorption curves might go some way towards contextualizing the observed behavior but it is going to be difficult to dissociate the temperature and light-dependent changes in gene expression. Finally, the link to human SAD is, I think, fairly tenuous and could be removed without impacting the science.

Response: We appreciate all of the comments of Reviewer 3 and have hopefully addressed them sufficiently by providing convincing data and explanations of our work.

We have (1) characterized the stimulus display system and lighting changes in Supplementary Fig. 1; (2) superimposed spectral data with absorption spectra for various photopigments in Supplementary Fig. 1; (3) dissociated the effects of temperature and photoperiod and discovered the crucial role of temperature in the seasonal regulation of the *LWS* gene.

We have attempted to mimic nature to the best of our ability, although it was difficult to match perfectly the natural orange-red color of real fish using 3D-CG. We are aware of the limitations of our 3D-CG system. In our experiments, we were able to compare the effect of red light from the monitor by changing the ratio of red light in the 3D-CG fish. How realistic the 3D-CG models are, or what colors Medaka actually see in the monitor, are difficult to determine, but do not take away from the significant behavioral differences we observed between wild-type and mutant fish. Taking all these factors into account, we are confident we performed our experiments under the best possible conditions and that our results are significant.

We have weakened the claim about the link to human SAD in the Discussion as follows: “This is consistent with our results showing significant gene expression changes in photopigments and the phototransduction cascade one week after transfer into LW conditions” has been changed to “It is interesting to note that we found significant gene expression changes in photopigments and the phototransduction cascade one week after transfer into LW conditions in Medaka fish.”

Reviewer #4

Seasonal changes are truly ubiquitous and cover virtually every aspect of life on this planet albeit 99% of the published work relates solely to reproduction. This fascinating paper from Yoshimura's group extends our knowledge into photoperiodic changes in color perception. As with all his work this contribution genuinely represents modern biological research "at the cutting edge": I note he is now using CRISP-R technology.

He has developed a new model system using Medaka fish that change their behavior from summer to winter and which can be reproduced experimentally by altering photoperiod. From the account offered it appears robust and quantifiable. Having established this platform he has then shifted fish to long days and at various times thereafter (days) he has analyzed retinal alterations in the families of opsin genes and in some of the downstream pathways. Many genes are altered (as one would expect) and he has focused for this paper on the so-called LWS genes. By using CRISP-R/CAS technologies he has altered expression of the LWS genes and produced statistically significant alterations in behavior.

I should have been proud to have offered this paper. Most importantly it is fascinating and interesting to a wide readership - what Nature is all about!. Secondly it mixes a range of techniques and approaches which represent the 21st Century.

PUBLISH IT!

Response: We are grateful to this reviewer for this wonderful comment.

Reviewer #1 (Remarks to the Author):

All my previous comments have been addressed. Congratulations on an excellent piece of work!

Reviewer #2 (Remarks to the Author):

In my opinion, the authors satisfactorily addressed all the points raised by the reviewers and adjusted the manuscript accordingly. Importantly, higher resolution pictures of in situ hybridization preparations on different parts of the retina, as well as additional data on the spectra used to create the white and red light environment, and on the spectral composition of the modeled fish are now provided in the supplementary information. Overall, the manuscript clearly improved in its quality and clarity, and I am happy to recommend it for being published.

It was a pleasure to review this study of visual ecology in Medaka,
Kind regards,
Sara Stieb

Reviewer #3 (Remarks to the Author):

The authors have made extensive revisions to this paper and have been extremely responsive to reviewer comments. This is very much appreciated.

The manuscript is much improved. I still, however, have two comments that I would like to see addressed:

1: Motion in cold vs warm tanks: Figure 1a clearly shows massive differences in mobility between the SL and LW conditions. Although the authors have now done a control experiment showing no differences in mobility, this result is hard to reconcile with their original data. I would like a lot more details on the nature of the control experiment (motion speed/distance in SL/LW conditions). In particular, I think I'd like to know how well that experiment matched the conditions that gave rise to 1b. Alternatively (as well?) the data in 1b could be shown as total time (as in 1d) rather than as % time....

2: Given that all the spectra are now provided, I think the authors should compute cone activations for the different stimuli. This is easy to do - effectively you just find the dot product of the photoreceptor absorbance spectra and the emission spectra. In this way, the relative activity and importance of the LWS and other photoreceptors would become very clear.

Point-by-point responses to Reviewer 3's comments to Shimmura *et al.*, "Dynamic plasticity in phototransduction regulates seasonal changes in color perception"

Below, the reviewer's comments are shown in blue, and our answers to his/her comments are shown in black.

Reviewer #3

The authors have made extensive revisions to this paper and have been extremely responsive to reviewer comments. This is very much appreciated.

The manuscript is much improved. I still, however, have two comments that I would like to see addressed:

1: Motion in cold vs warm tanks: Figure 1a clearly shows massive differences in mobility between the SL and LW conditions. Although the authors have now done a control experiment showing no differences in mobility, this result is hard to reconcile with their original data. I would like a lot more details on the nature of the control experiment (motion speed/distance in SL/LW conditions). In particular, I think I'd like to know how well that experiment matched the conditions that gave rise to 1b. Alternatively (as well?) the data in 1b could be shown as total time (as in 1d) rather than as % time....

Response:

Reviewer 3's concern is how well the data shown in Supplementary Figure 2 matches the experiment shown in the original Figure 1b. Actually, we did not perform a new experiment, but simply re-analyzed the original data from Figure 1b to calculate the swimming speed/distance, which is shown in Supplementary Figure 2. We apologize that this point was not more clearly stated. We have included this information in the legend of Supplementary Figure 2 as follows: "Swimming distance and speed were calculated from the data shown in Figure 1b."

2: Given that all the spectra are now provided, I think the authors should compute cone activations for the different stimuli. This is easy to do - effectively you just find the dot product of the photoreceptor absorbance spectra and the emission spectra. In this way, the relative activity and importance of the LWS and other photoreceptors would become very clear.

Response

We appreciate this additional comment; however, it is already quite clear from Supplementary Figure 1 that LWS is the primary photopigment responsible for detection of long wavelength light, including the orange-red coloration of real/virtual fish and the monochromatic red light

stimulus. We believe that absorbance spectra are not sufficient to calculate relative activity or importance of each photopigment. For example, to do this accurately, we should take into account both the sensitivity and the exact number of cones. Unfortunately, this information is currently not available for Medaka fish. Without this crucial information, the calculations would be inaccurate and might be misleading. Therefore, we believe it is better not to include this information in our manuscript.

Reviewer #3 (Remarks to the Author):

1: Swimming distance: The data from Figure 1b are interesting - given that there was a clear hypothesis about swimming speed, I assume the t-tests are 1-sided?

But really I wanted to see the same data for figure 1a. There seems to be almost no chance that the distances are the same - and therefore the paper should contain a sentence saying something like "In general, fish swim very much faster, and further, (by a factor of xx) in the LW compared to the SL conditions". Subsequent data should be interpreted with this in mind...

2: "Clear that LWS is the primary photopigment..." : It's not clear at all - for example, the main difference between the orange and gray models occurs at two points in the spectrum. One (around 610nm) is probably picked up by the LWSA/B receptor. The other (around 450nm) is clearly not. If we really don't know anything about the numbers of different photoreceptors in the eye (really?) it's therefore entirely possible that the LWSA/B could be expressed only very sparsely and have absolutely nothing to do with distinguishing the orange models from the gray controls. This should be mentioned somewhere...

Point-by-point responses to Reviewer 3's comments to Shimmura *et al.*, "Dynamic plasticity in phototransduction regulates seasonal changes in color perception"

Below, the reviewer's comments are shown in blue, and our answers to his/her comments are shown in black.

Reviewer #3

1: Swimming distance: The data from Figure 1b are interesting - given that there was a clear hypothesis about swimming speed, I assume the t-tests are 1-sided?

Response:

The two-tailed *t*-test was performed. However, the results of one- and two-tailed tests were the same.

But really I wanted to see the same data for figure 1a. There seems to be almost no chance that the distances are the same - and therefore the paper should contain a sentence saying something like "In general, fish swim very much faster, and further, (by a factor of xx) in the LW compared to the SL conditions". Subsequent data should be interpreted with this in mind...

Response:

We have provided the swimming distance and speed data in the supplementary figure and added the following explanation in the text as suggested.

"In general, fish swim much faster and further in LW compared to SC conditions (Supplementary Fig. 1a)."

2: "Clear that LWS is the primary photopigment..." : It's not clear at all - for example, the main difference between the orange and gray models occurs at two points in the spectrum. One (around 610nm) is probably picked up by the LWSA/B receptor. The other (around 450nm) is clearly not.

Response:

As written in our previous rebuttal letter, it has been demonstrated that color cue is perceived as the relative balance of brightness at multiple wavelengths rather than the brightness at a specific wavelength in mate discrimination in medaka. Although multiple photopigments are required to perceive "color", our results clearly demonstrated that LWS is indeed necessary to discriminate orange-red body coloration.

If we really don't know anything about the numbers of different photoreceptors in the

eye (really?) it's therefore entirely possible that the LWSA/B could be expressed only very sparsely and have absolutely nothing to do with distinguishing the orange models from the gray controls. This should be mentioned somewhere...

Response:

The exact numbers of each photoreceptor are unavailable in the medaka eye. However, as shown in Figure 2 and Supplementary Figure 6, LWS opsins are indeed expressed in whole retina. This information is already described in the text.